# A Cross Sectional Study Evaluating Psychosocial Job Stress and Health Risk in Emergency Department Nurses

**DOI:** 10.3390/ijerph16183243

**Published:** 2019-09-04

**Authors:** Rupkatha Bardhan, Karen Heaton, Melissa Davis, Peter Chen, Dale A. Dickinson, Claudiu T. Lungu

**Affiliations:** 1Department of Occupational Safety and Health, Murray State University, Murray, KY 42071, USA; 2School of Nursing, University of Alabama at Birmingham, Birmingham, AL 35294, USA; 3Department of Environmental Health Sciences, University of Alabama at Birmingham, Birmingham, AL 35294, USA (M.D.) (D.A.D.); 4Department of Psychology, Auburn University, Auburn, AL 36849, USA

**Keywords:** nurses, job stress, oxidative stress, effort–reward imbalance, lipid peroxidation

## Abstract

Nurses experience psychosocial work stress that may negatively affect physical and mental health over time. In this cross-sectional study we investigated prevalence of job stress and oxidative stress in nurses, and determined if significant relationships exist between higher job stress scores and demographic factors and working conditions. Emergency department nurses (*n* = 42) were recruited from a University Hospital following Institutional Review Board approval. Job stress indicators, effort–reward ratio and overcommitment were evaluated from survey questionnaires using the effort–reward imbalance model, and associations with age, sex, body mass index, and working conditions were measured by logistic regression analysis. Oxidative stress biomarkers, 8-isoprostane, malondialdehyde, and antioxidant levels were measured from urine specimens. Job stress was prevalent with effort–reward ratio > 1 in 93% and overcommitment > 50 in 83% of the study participants. Age, body mass index, years of experience, weekend work, work hours per week, and shift work showed strong associations with effort–reward ratio and overcommitment scores. Malondialdehyde was higher in participants with high overcommitment. We report that psychosocial job stress is prevalent among nurses, as revealed by the high effort–reward and overcommitment scores. Job stress may be reduced through implementation of appropriate stress reduction interventions.

## 1. Introduction

Nurses in the emergency departments (ED) work in a highly stressful environment, which includes but is not limited to long work hours, handling life threatening diseases, interacting with accident and trauma victims, dealing with anxious patient families, and exacting superiors. Such intense job demands often lead to declines in physical and mental health in medical personnel [1,2]. Studies have identified high levels of stress and suicide among nurses [3,4]. In a cross-sectional study conducted on 850 participants from Hong Kong, it was reported that nurses were more depressed, anxious, and stressed than the general local population [5].

Nursing is a rewarding career path as nurses save lives, but is paradoxically considered one of the most stressful professions. Nurses can face physical and emotional challenges every day at work, which is often compounded by factors such as understaffing due to cost-cutting, increasing workload, and rise in the use of sophisticated healthcare technologies [6]. Over the past decade, nursing has consistently ranked at the highest level regarding occupational stress among all healthcare professions [7]. Surveys consistently found nurses are stressed and overworked, and reported concern about their safety and health [8]. While the US provides one of the best working conditions for nurses regarding full-time employment opportunity and pay, reports of stress and burnout are quite common [9,10].

Job stress has been relatively neglected by psychologists and researchers until recent years [11]. Stressed employees are vulnerable to injury and ill-health, leading to absenteeism and loss of productivity [12]. Corrective measures need to be implemented at workplaces to reduce job-related stress among employees to minimize these losses.

Psychological stress has been implicated in the elevated production of reactive oxygen species (ROS) causing oxidative stress [13,14,15,16]. Oxidative stress (OS) has been identified as a major risk factor for cardiovascular diseases (CVD) and cancer [17,18,19,20]. Job stress has been recognized as one of the major risk factors for coronary heart disease (CHD) [21,22]. A study involving industry workers in Japan found significant association between job stress and oxidative stress biomarkers [23].

Oxidative stress results from an imbalance between the production of oxygen radicals and the ability of the body to remove or detoxify their harmful effects through neutralization or reduction by antioxidants [24]. Lipid peroxidation is the oxidative degradation of lipids in which free radicals remove electrons from the lipids in cell membranes, resulting in cellular damage, which may increase risk for DNA mutations and carcinogenesis [25]. Malondialdehyde (MDA) and 8-isoprostane (8-IsoP) are well-known biomarkers of oxidative stress-mediated lipid peroxidation [24,26] and are routinely used to demonstrate oxidative stress in biological specimens [27].

A strong association between working schedules of participants and psychological stress is well documented. Working long hours and night shifts were shown to affect biological rhythms, increasing health risks in employees [28,29]. Shift workers with and without night shifts more often have difficulties combining work and life than do those who only work the day shift [30,31,32]. Unfavorable working hour characteristics including long work weeks, evening and night shifts, weekend work, and quick returns are associated with work–life balance conflict and increased chance of work-related injury [33,34].

Psychosocial stress is positively associated with BMI and increased weight gain [35]. The aim of our present study was to determine whether psychosocial job stress exists among ED nurses in the US, and further investigate any relationship between psychosocial job stress and demographic factors, work conditions, and oxidative stress in this workforce. To the best of our knowledge, this is the first study which evaluated the relationship between psychosocial job stress and oxidative stress/lipid peroxidation in an ED nurse workforce in a US non-profit hospital.

## 2. Materials and Methods

### 2.1. Study Design

A cross sectional pilot study was conducted to investigate the impact of psychosocial stress on ED nurses in a US non-profit hospital. Between September 2015 and February 2017, nurses working in an emergency department in a US hospital participated in our study. All the nursing staff that worked in the ED (*n* = 50) were identified and approached for the study. A total of 42 participants (*n* = 42) agreed to participate in our study with a response rate of 84% (42/50) (Figure 1). Both men and women nurses of different racial and ethnic backgrounds participated in the study.

Selection criteria for participants included nurses working in the emergency department, above 19 years of age, and with no physical and mental disorders. The exclusion criteria for the study were participants below age 19 years, physically or mentally challenged, and pregnant women. There were no participants that worked in the ED who fit the exclusion criteria. Recruitment of ED nurses were conducted by posting flyers and directly talking to ED nurses during their staff meetings. The hospital nursing administration supported the study, and Institutional Review Board (IRB) approval was obtained prior to recruitment. All subjects gave their informed consent for inclusion before they participated in the study. Written and signed informed consent was obtained from all participants. The study was conducted in accordance with the Declaration of Helsinki, and the protocol was approved by the Ethics Committee of University of Alabama at Birmingham (Protocol # 150625004). Participants filled out the effort–reward imbalance questionnaire. Participants also provided information about their work schedule including work hours per week, whether they work night shifts and weekends, and their years of work experience. Age, body mass index (BMI), and sex, were also provided by participants in the survey, and were included in the analyses as covariates. After the survey, spot urine samples were collected in sterile tubes from the participants, centrifuged at high speed to remove any cell debris and insoluble particles (10,000× *g* for 5 min), and supernatants were immediately aliquoted into multiple tubes and stored in −80 °C for future analysis.

### 2.2. Evaluation of Job Stress Using the Effort–Reward Imbalance Model

Job stress was calculated from the responses given by the participants in the survey questionnaires based on effort–reward ratio based on effort–reward imbalance (ERI) model, developed by Johannes Siegrist [1,2]. The short version of the ERI model was utilized. The ERI model consists of two main scales: extrinsic effort and reward. Effort and reward were measured using a 4-point response option. Two job stress indicators were identified: effort–reward ratio (ER Ratio) and overcommitment (OC). Effort measures the amount of effort a person puts in his work and was measured using a 4-point Likert scale (strongly disagree, disagree, strongly agree, and agree). Effort questionnaires were coded as -, ERI 1 to 3, with 4-point Likert scale, - giving a total score between 3 and 12. Questions on effort were focused on the following topics: constant time pressure due to heavy workload, interruptions and disturbances while performing work, job becoming more demanding over the past years. Reward was measured using the 4-point Likert scale with 7 questions coded as ERI 4 to 10, a sum score of which varies between 7 and 28 (Figure 2). The lower the score, the fewer occupational rewards were received by the person. Questions on reward were focused on receiving respect from superiors, job promotion prospects, job security, and adequate salary. The OC questions captured the essence of personal pattern of coping with work. OC was measured by 6 items coded as overcommitment imbalance, or OC 1 to 6, with 4-point Likert scale, and total score could vary between 6 and 24 (Figure 2). Questions on OC were focused on topics like overwhelmed by time pressures, relaxed and switched off from work at home, sacrificing too much for job, and trouble sleeping at night due to concerns over incomplete or undone work. Psychometric properties of the ERI questionnaires are shown in Figure 2. The ER ratio (ERR) is the balance between effort and reward scales. ERR captures the imbalance between effort and reward at the individual level. To compute the ER ratio, effort (E) was in the numerator and reward (R) was in the denominator, and K was the correction factor that was used to adjust for the unequal number of the effort and reward scores. The ER ratio was calculated by the formula, ERR = K (x) × (E/R), where K = 7/3 correction factor proposed by the ERI model was used in this study. For ER = 1 the person reports one effort for one reward, for ER < 1 the person reports less effort for each reward, for ER > 1 the person reports more effort for each reward [3,4,5]. OC was divided into high (OC > 50) and low (OC ≤ 50) groups with a cut point of 50%. There is no specific cut point for overcommitment mentioned by Siegrist et al. in the effort–reward imbalance model. Scores above 50% on the overcommitment scale were characterized as high and scores below 50% were characterized as low.

### 2.3. Evaluation of Oxidative Stress/Lipid Peroxidation from Urine

Lipid peroxidation biomarkers, MDA, and 8-IsoP were measured from urine samples of participants using quantitative assays described below. Total antioxidant activity, which represents oxidative stress, was also determined from the urine samples. All the standards and the samples were assayed in duplicate and intra- and inter-assay coefficient of variation was found to be less than 5%. MDA, 8-IsoP, and total antioxidant values were normalized with urinary creatinine concentrations.

#### 2.3.1. Urinary Malondialdehyde Assay

MDA, was measured in the urine samples by a thiobarbituric acid reactive substances (TBARS) assay using a commercially available kit (OxiselectTM, Cell Biolabs, Inc, San Diego, CA, USA, Cat # STA 330) following the manufacturer’s instructions. Prior to the assay, urine samples were thawed and centrifuged at 10,000× *g* for 5 min to remove insoluble particles. Supernatants were collected and assayed. All standards and urine samples were assayed in duplicate. One hundred microliters (100 µL) of each sample and standard were added to 1.5 mL sterile micro centrifuge tubes. A total of 100 µL of sodium dodecyl sulfate (SDS, supplied with the kit) detergent solution was added to the tubes and incubated for 5 min at room temperature. Two hundred and fifty microliters of thiobarbituric acid (TBA) reagent was then added and the tubes were incubated at 95°C for 50 min in heating blocks (or water bath). The tubes were then cooled in an ice bath for 5 min, and centrifuged at 3000 rpm for 15 min. A total of 200 µL of the supernatants from the samples and standards were transferred to 96-well microplates and absorbance was read at 532 nm using a microplate reader (Biotek Instruments, Winooski, VT, USA). Sample micro-molar (µM) concentration was calculated from the linear regression of the standard curve by subtracting the y-intercept value of the standard curve linear equation from sample average absorbance values and divided by the slope of the standard curve linear equation slope value, multiplied by dilution factor.

#### 2.3.2. Urinary 8-Isoprostane Assay

Participants’ urinary 8-IsoP, was measured via enzyme-linked immunosorbent assay (ELISA) using a commercially available kit (Cayman Chemical, Ann Arbor, MI, USA; Cat # 516351) following the manufacturer’s protocol. This assay was based on competitive binding of 8-IsoP to an antibody against 8-IsoP over a known concentration of 8-IsoP-acetylcholine esterase (tracer). Briefly, 50 μL of each of the samples and standards, tracer, and 8-IsoP antibody was added in 96-well microplates pre-coated with mouse anti-rabbit IgG. The microplates were incubated for 18 h at 4°C. The next day, wells were washed with phosphate buffered saline (PBS) and acetylcholine esterase activities were measured adding Ellman’s substrate reagent (a mixture of acetylcholine and 5, 5’-dithio-bis- (2-nitrobenzoic acid) to the wells. Colorimetric determination of acetylcholine esterase activity was performed by measuring absorbance at 415 nm using a microplate reader (Biotek-,Winsooksi, VT, USA) following hydrolysis of acetylcholine by acetylcholine esterase generating a yellow-colored compound, thiocholine. Unknown sample concentrations (picograms/mL; pg/mL) of 8-IsoP were determined by calculating the amount of free 8-IsoP present in the wells against a standard curve generated by plotting absorbance values of acetylcholine esterase-bound 8-IsoP.

#### 2.3.3. Total Antioxidant Capacity Assay

Total antioxidant capacity was measured in the urine specimens using a commercially available antioxidant assay kit (Cayman Chemical Cat # 709001) following the manufacturer’s instructions. Briefly, urine samples were diluted 1:10 with assay buffer. Ten microliters (10 μL) of sample/standards, 10 μL of met-myoglobin, and 150 μL of chromogen was added to the 96 well plates. The reactions were initiated by 40 μL of hydrogen peroxide working solution and incubated on a shaker for 5 min at room temperature. The absorbance was read at 750 and 405 nm using a microplate reader. Antioxidant concentration (µM) was calculated from the linear regression of the standard curve by subtracting the y intercept value of the standard curve linear equation from sample average absorbance values and divided by the slope of the standard curve linear equation slope value, and multiplied by dilution factor.

#### 2.3.4. Creatinine Assay

Spot urine samples were adjusted by creatinine to normalize for urine analyte concentration [6]. Creatinine was assayed by a commercially available kit (Cayman Chemical, Cat # 500701) following the manufacturer’s protocol. Urine was diluted 1:10 with HPLC-grade water before the assay. A total of 15 μL of each of the standards and samples were added to 96-well plates, and reactions were initiated by adding alkaline picrate solutions to all the wells. The plates were incubated in a shaker at room temperature for 10 min and absorbance was read at 495 nm using a microplate reader. Five microliters of acid solution was added to the wells and incubated on a shaker for 20 min at room temperature and a second absorbance was read at 495 nm. The average initial and final absorbance of samples and standards were calculated. The average final absorbance was subtracted from the average initial absorbance to get the average corrected absorbance. Then, adjusted absorbance was calculated by subtracting average corrected absorbance of standards from itself and all other samples and standards. Adjusted absorbance was plotted in the standard curve and linear regression equation was determined. Creatinine concentration was measured in milligrams/deciliter (mg/dL) from the linear regression equation of the standard curve by subtracting the y-intercept value of the standard curve linear equation from sample absorbance values and divided by the slope of the standard curve linear equation slope value, multiplied by dilution factor.

### 2.4. Statistical Analysis

All analyses were performed by SAS 9.4 and Graph Pad Prism Software Windows version 7. Two types of statistical analyses were performed. Descriptive statistics: Percentage (%), mean, standard deviation (SD), median, range; and analytic statistics: Wilcoxon–Mann–Whitney test was performed which is a non-parametric test of significance. Lipid peroxidation biomarkers (MDA. 8-isoprostane, and antioxidants) were compared between two job stress indicator study groups: High vs. low ERR and OC (ERR > 1 vs. ERR ≤ 1, and OC > 50 vs. OC ≤ 50) using the Wilcoxon–Mann–Whitney test. Data distributions were tested for normality using the Shapiro–Wilk test. Binary logistic regression analysis was performed to examine the independent effects of the age, BMI, sex, years of experience, weekend work, work hours, and shift work on job stress indicators, ERR and OC (all variables categorical) (Figure 3). The results were shown as odds ratio and 95% confidence interval. Spearman’s rank correlation coefficient (rho) was used to study the association between job stress indicators, ERR and OC (independent variables, continuous); and lipid peroxidation biomarkers, MDA, 8-isoprostane, and antioxidants (dependent variables, continuous). Statistical significance was accepted at the 5% level.

## 3. Results

### 3.1. Participant Characteristics

Nurses’ personal and occupational characteristics are reported in Table 1. Median age of the study participants was 30 years. Sixty-nine percent of the study participants were females, and 31% were males. BMI was also calculated from self-reported individual weight and height, and median BMI was found to be 27.3. Fifty percent of the study participants were married. The majority of the participants (79%) had spent 10 or fewer years on the job. Rotating shifts (day and night shifts) were more common (55%). Fifty percent of the nurses reported working a >40 h work-week, while 95% worked exclusively on the weekends.

### 3.2. Potential Health Risk Indicators Measured in This Study

Job stress was measured by calculating the effort–reward ratio (ERR; high effort, low reward), and capacity to cope with high demand at work (OC). Ninety-three percent of the nurses in the study reported high levels of stress (ERR > 1), and 83% were above a cut-off OC score of 50 (Table 2). Biomarkers for oxidative stress were also measured from the spot-urine samples. Average concentrations of MDA and 8-IsoP were determined to be 19.34 μM (median 19.35) and 1452.9 pg/mL (median 349.68), respectively. Average antioxidant levels were found to be 480.2 μM (median 480.4). Data are reported in Table 2.

### 3.3. Distribution of the Statistical Variables

Normality testing (Shapiro–Wilk test) was performed to analyze the distribution of each of the study variables (health risk indicators) so that proper statistical methods could be applied to determine association between the variables (parametric or non-parametric), if any. Results of the analyses showed ERR and OC followed a normal distribution, whereas oxidative stress (MDA, 8-IsoP, and antioxidants) were not normally distributed (Table 3).

### 3.4. Association between Job Stress Indicators (ERR and OC) and Oxidative Stress Biomarkers

Non-parametric Spearman rank correlation was used to determine association between the job stress indicators and oxidative stress biomarkers in the urine. Results indicated no significant correlation (*p* > 0.05) between ERR and any of the oxidative stress biomarkers (MDA, 8-IsoP, antioxidant) (Table 4). Similarly, no significant correlation (*p* > 0.05) was found between OC and any of the oxidative stress biomarkers (Table 5). Non-parametric Wilcoxon–Mann–Whitney test was performed to determine if there is difference in oxidative stress between nurses with high (>1; -*n* = 39) and low ERR (≤1; -*n* = 3). Results showed MDA, 8-IsoP, and antioxidant levels were equivalent between the high and low ERR groups (Table 6). A similar test was performed to determine difference in the levels of oxidative stress biomarkers between nurses with high (>50) and low OC ≤ 50) score. Data showed that there was a significant difference in MDA levels (Test Statistic = 89, *p* = 0.03) between nurses with high and low OC (Table 7). Spearman rank correlation was tested to find positive correlation between MDA and OC (Rho 0.13, *p* = 0.40), although not significant (Figure 4). Antioxidant and 8-IsoP levels were equivalent between the high and low OC groups (Table 7). Although there were differences between ERR and OC regarding their individual relationship with oxidative stress biomarkers (MDA in this case), ERR and OC were positively correlated (Pearson correlation, r = 0.38; *p* < 0.01) (Figure 5). Significant difference (*p* < 0.04) was observed in MDA levels between high and low overcommitted participants (Figure 6). Females working night shift reported significantly lower 8-IsoP levels compared to females working day shift (*p <* 0.001) using Mann–Whitney test (Figure 7).

### 3.5. Job Stress as a Function of Age, BMI, and Work Conditions

Logistics regression analysis was performed to determine whether nurses’ age, BMI, and work conditions are risk factors of job stress. BMI (obese: OR = 2.653, CI = 0.026–34.15; overweight: OR = 1.731, CI = 0.208–14.39), less or equal to 10 years at the job (OR = 1.976, CI = 0.05–69.20), and weekend work (OR = 2.913, CI = 0.033–254.75) were found to be potential risk factors for high ERR (>1). Age over 40 years (OR = 0.423, CI = 0.005–33.97), being female (OR = 0.640, CI = 0.051–7.95), and rotating shift (OR = 0.309, CI = 0.031–3.09) appeared to lower the risk of high ERR (Table 8).

BMI (obese: OR = 2.006, CI = 0.226–17.77; overweight: OR = 1.684, CI = 0.223–12.727), age (>30–40: OR = 1.903, CI = 0.111–32.70; >40: OR = 4.143, CI = 0.107–160.47), being female (OR = 2.967, CI = 0.397–22.18), working rotating shift (OR = 2.683, CI = 0.45–15.94), weekend work (OR = 2.653, CI = 0.020–115.13), and working more than 40 h per week (OR = 2.429, CI = 0.37–15.56) showed strong associations with high OC (>50). Whereas more than 10 years of work experience (OR = 0.177, CI = 0.009–3.57) demonstrated low risk towards the development of high OC (Table 9).

## 4. Discussion

Nurses work at the forefront of the healthcare industry and perform a highly demanding job, predisposing them to a great deal of psychosocial job stress compared to many other professions. Exposure to high levels of stress at work poses a significant health risk among nurses in later life. Several studies have demonstrated elevated stress levels in nurses [7,8]. Chronic oxidative stress-induced cellular damage has been implicated in many debilitating diseases such as heart disease and cancer [9]. A few studies conducted outside the United States (US) showed significant relationship between job stress and biological markers of oxidative stress in nurses [10]. Lipid peroxidation, a critical biomarker of oxidative stress, is a well-known risk factor for many life-threatening diseases, including heart disease and cancer [11,12,13,14,15,16]. The aim of the present study was to determine if oxidative stress is prevalent among nurses with higher psychosocial job stress (ERR > 1 or OC > 50). We further evaluated if higher BMI, age, and work schedules contributed to elevated job stress in the participants, thus increasing long-term health risk in ED nurses.

The primary findings from the study demonstrated high levels of stress in the study nurses. The ERI questionnaire was used in this study to measure psychosocial job stress. The ERI model is a validated approach to measure psychosocial work stress by identifying non-reciprocity between occupational efforts spent and rewards received, and has the ability to predict poor health outcomes [17]. ERI and OC were shown to be strongly associated with depressive episodes at work in Brazilian and New Zealand studies [18,19]. The ERI model was also successfully applied to measure occupational stress and burnouts in physicians and nurses working in intensive care units [20,21]. The popularity of the ERI model for screening psychosocial stress at work may be understood by the fact that the questionnaire has been used in many other healthcare settings and translated in several languages [17,22]. The effort–reward imbalance model states that mismatch between high effort and low reward may lead to adverse health outcomes [1,2,4]. An ER ratio greater than 1 was observed in 93% (39/42) of the participants in the study, indicating psychosocial stress. Similarly, the overcommitment hypothesis states that having a high level of personal commitment may also result in reduced health even in the absence of effort–reward imbalance. Our data showed the presence of high OC score (>50) in 83% (35/42) of the participants. High ERR and OC scores suggests that ED nurses are at a potential risk of developing stress-related disorders in later life. Interestingly, our data suggested that effort–reward imbalance was associated with male participants, while female participants were more overcommitted to their work. A study carried out in Germany reported similar observation among males, while no gender difference was found for overcommitment [23].

Our data showed significant association between high BMI and psychosocial job stress. In a National Population Health Survey in Canada, men (but not women) who worked more than 35 h a week tended to be more overweight. In a Nurses Health Study in the United States, higher job strain was also linked to higher BMI [24,25]. Age of participants also showed a positive relationship with job stress in our study, which was supported by other observations [26,27]. The majority of the participants (79%) in our study worked 10 or fewer years on the job. Most of the nurses in our study were also young (mean age 33). Research indicated critical care nurses perceive higher psychological stress during the first two years of their employment [28]. It may be assumed that more experienced nurses will effectively handle stress at work. But data suggests increased incidence of stress may occur in older individuals due to age-related burn out and lower physical ability to perform at the highest level [29]. Inexperienced nurses may benefit from their more experienced peers in reducing stress at work and preserving health. 

Our data indicated that female nurses demonstrated more overcommitment at work, while male nurses reported higher effort–reward imbalance. MDA and 8-IsoP, well known biological markers of oxidative stress-induced lipid peroxidation [30,31,32,33], showed no significant correlation with either ERR or OC. There was no significant difference between MDA levels between nurses with high and low ERR, whereas significantly higher levels of MDA were observed in nurses with higher OC (>50). This suggests that overcommitment may be a better predictor for oxidative stress compared to effort–reward imbalance. Among the female nurses, 8-IsoP levels were also lower in nurses working night shift (shift work) over those who worked only the day shift. This is in line with our data showing that shift work was associated with lower ERR. Total antioxidant capacities were within the normal range suggesting overall oxidative stress may not be at abnormal levels in the study participants. A few previous studies reported that a significant relationship exists between occupational stress and oxidative stress biomarkers in nurses [10,34]. In one study, performed on 204 registered nurses from two hospitals in Egypt, significant associations were noted between job stress indicators (ER ratio and OC) and oxidative biomarkers (MDA, antioxidant capacity) in the blood of nurses with more than 10 years of work experience [34]. Another study, carried out on nurses in China reported significant association between occupational stress, oxidative stress, and reduced antioxidant capacity in the nurses with more than 25 years of experience [10]. The current study supports previous study findings in that significant associations were observed between the job stress indicator, OC, and the lipid peroxidation biomarker MDA. No significant relationship was observed between job stress indicators and 8-isoprostane (8-IsoP).

Shift work, working weekends, and long working hours have been proven to be contributors to health risk. Working night shift and aging were reported to be the contributing factors for burn-out syndrome in a group of intensive care unit nurses in Spain, accompanied by elevated oxidative stress biomarkers [34]. Our study showed high ERR was associated with weekend work, whereas working rotating shift (compared to only day shift), weekend work, and working over 40 h/week were associated with high overcommitment. Night shift more than day shift was found to be a potential risk factor for the development of oxidative stress among nurses in Poland and Turkey [35,36,37]. In our study, 8-IsoP was significantly lower in females working night shifts than day shifts. Lower 8-IsoP levels in the female nurses working night shift could be attributed to high estrogen production during night-time sleep-off cycle. The sleep hormone, melatonin, stimulates estrogen release, which might play a protective role against oxidative stress in premenopausal females working night shifts [37]. Although high odds ratios (relative risk) were observed to define the relationship between dependent (ERR and OC) and the independent variables (age, BMI, work conditions, oxidative stress biomarkers), none of those relationships were found to be statistically significant. Low sample size may have contributed to this low level of significance.

In summary, psychosocial job stress was observed in most participants in our study. All efforts were made to minimize selection bias of the study by approaching all the participants who worked in the emergency department, with male and female nurses of different race, BMI, and age participating in our study. There were several reasons a cross sectional study was chosen; the exploratory study gave the opportunity to investigate stress at a specific point of time and provided opportunity for future research, besides being affordable and less time consuming. This study had several limitations. First, it is a cross sectional study design, so we were unable to demonstrate any chronobiological effects of work stressors. We could not determine any information on the causality of the job stress indicators and oxidative stress biomarkers. Follow-up investigations over decades among these participants will inform any change in relationship between job stress indicators and oxidative stress due to aging. In addition, the relatively small sample size may have limited the generalizability of the study. In the low effort–reward ratio group (ERR ≤ 1) there were only three participants, compared to 39 participants in the high ERR group (ERR > 1). In the low over-commitment group there were only seven participants (OC ≤ 50), compared to 35 participants in the high overcommitted group (OC > 50). In the future, more participants from different hospitals could be recruited to expand this study. Multiple hospital emergency departments could be studied for identifying associations between psychosocial job stress indicators and lipid peroxidation biomarkers.

Autonomic system dysfunction, hypothalamic pituitary axis assessment, and heart rate variability are early predictors of cardiac risk and are associated with oxidative stress [38,39]. Collecting data on such parameters in future studies will help in further understanding of the mechanism linking occupational stress with health risks in nurses. Nurses are commonly exposed to long work hours as well as shift work. Such demanding schedules often lead to sleep disorders because of the need to sleep at irregular times, resulting in irregular circadian rhythms, poor sleep quality, and shorter sleep duration [40]. Epidemiological studies support increased risk of cardiovascular diseases among shift workers suffering from sleep disturbances [41,42]. Future investigations testing the relationship between ERI and sleep pattern among nurses working night shifts will be immensely helpful in connecting job stress with the risk of cardiovascular diseases. Despite these limitations, the findings of the study reflect the association between the psychosocial work environment and oxidative stress in the nursing job. Therefore, these results should be verified with large-scale and multi-center studies. Several studies have focused on the role of effective interventions to reduce stress among nurses. The practice of yoga and mindfulness have been shown to improve the physical, emotional, and mental health of healthcare workers via reduction of stress and burnout [43,44]. A Canadian study has demonstrated that the presence of authentic leadership, organizational identification, and occupational coping self-efficacy in the workplace have positive outcomes on job turnover intentions of new graduate nurses [45]. Implementing effective interventions in hospitals to reduce stress among nurses can be beneficial to both nurses and their patients.

Although several limitations exist in our study, the study is significant in understanding the psychosocial work environment in a hospital emergency department and possible associations of psychosocial job stress and health risk factors in ED nurses.

## 5. Conclusions

In our study, we found that the psychosocial work environment may be associated with occupational stress among nurses, as revealed by high ERR, OC, and elevated MDA levels. Overall, this study reported (based on high effort–reward and overcommitment scores) that emergency department nurses working in a US non-profit hospital experience psychosocial stress at work. However, a longitudinal study involving more hospitals may be conducted to strengthen the association between psychosocial work stress and oxidative stress in emergency department nurses.

## Figures and Tables

**Figure 1 ijerph-16-03243-f001:**
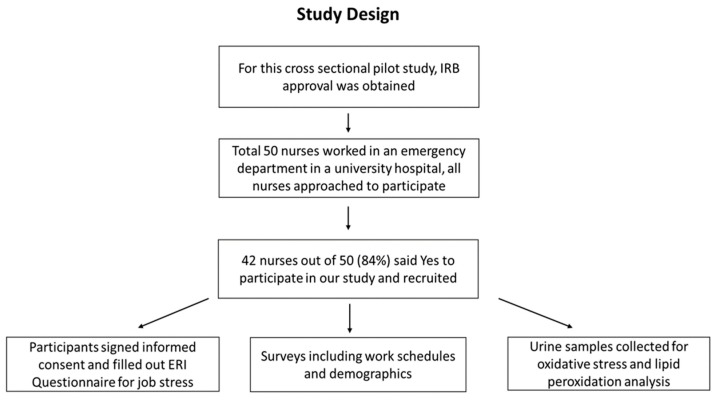
A flow diagram showing the study design. This study design was adopted following Institutional Review Board (IRB) approval to investigate the impact of psychosocial stress on emergency department nurses in a United States hospital.

**Figure 2 ijerph-16-03243-f002:**
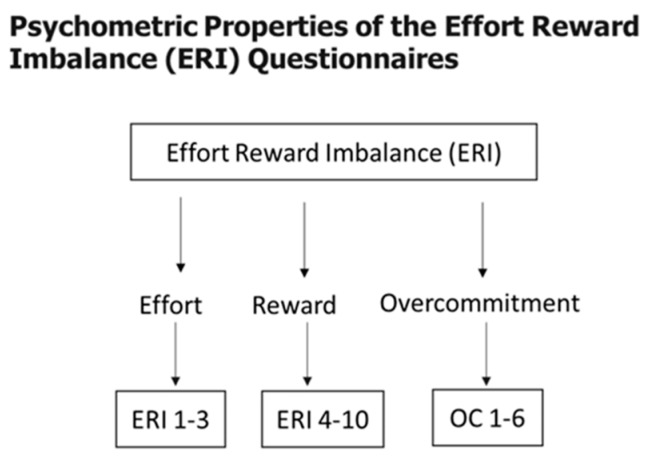
Psychometric properties showing effort–reward imbalance (ERI) questionnaires. Effort–reward imbalance survey questionnaires included three effort questions (1–3), seven reward questions (4–10), and six overcommitment questions (1–6).

**Figure 3 ijerph-16-03243-f003:**
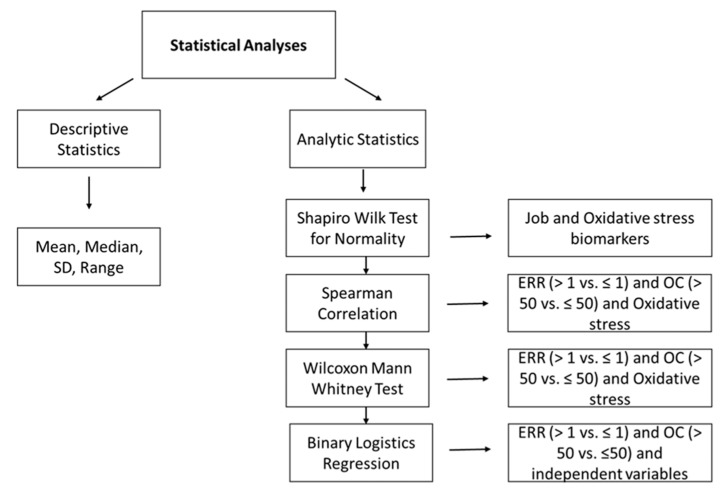
Flow diagram showing rationale for various statistical methods used for data analysis. Two types of statistical analyses were performed: Descriptive statistics for computing percentage (%), mean, standard deviation (SD), median, range of variables; and analytical statistics: Shapiro–Wilk test was performed to determine normality, Wilcoxon–Mann–Whitney test was used to analyze non-parametric data, and binary logistic regression analysis was performed to measure association between job stress indicators, and independent variables ERR: Effort Reward Ratio; OC: Overcommitment.

**Figure 4 ijerph-16-03243-f004:**
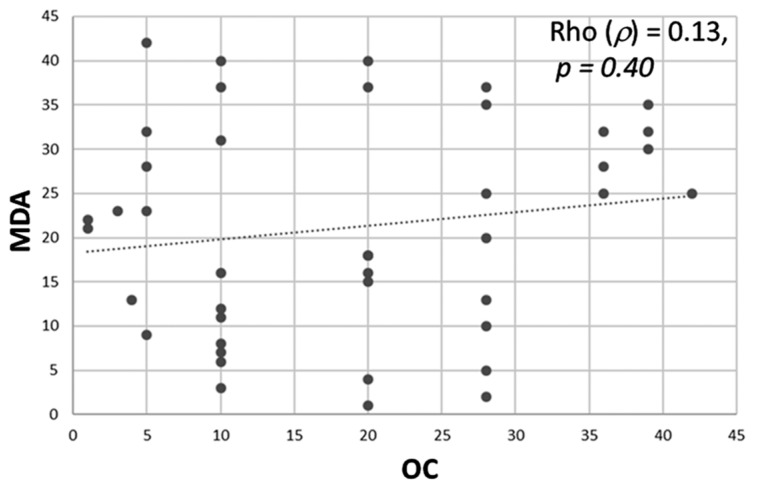
Scatter plot showing correlation between overcommitment (OC) and oxidative stress biomarker, malondialdehyde (MDA). OC was determined from the effort–reward imbalance (ERI) questionnaires filled out by study participants (*n* = 42). Spearman rank correlation showing positive correlation between OC vs. MDA (Rho = 0.13, *p* = 0.40).

**Figure 5 ijerph-16-03243-f005:**
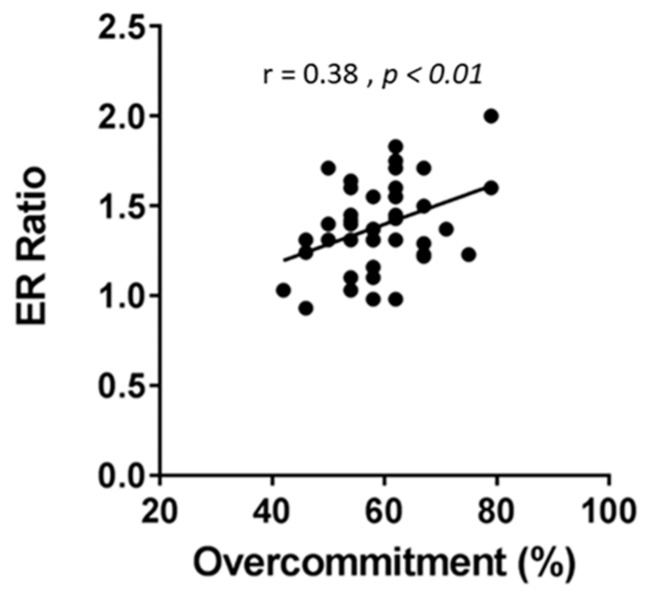
Scatter plot showing correlation between job stress indicators, effort–reward ratio, and overcommitment. Job stress (ER ratio, and OC) was determined from the effort–reward imbalance (ERI) questionnaires filled out by study participants (*n* = 42). ERR and OC scores were found to be normally distributed. Pearson r correlation was used to define relationship between ERR and OC scores (Pearson r correlation = 0.38, 95% CI 0.087–0.614, *p* = 0.01).

**Figure 6 ijerph-16-03243-f006:**
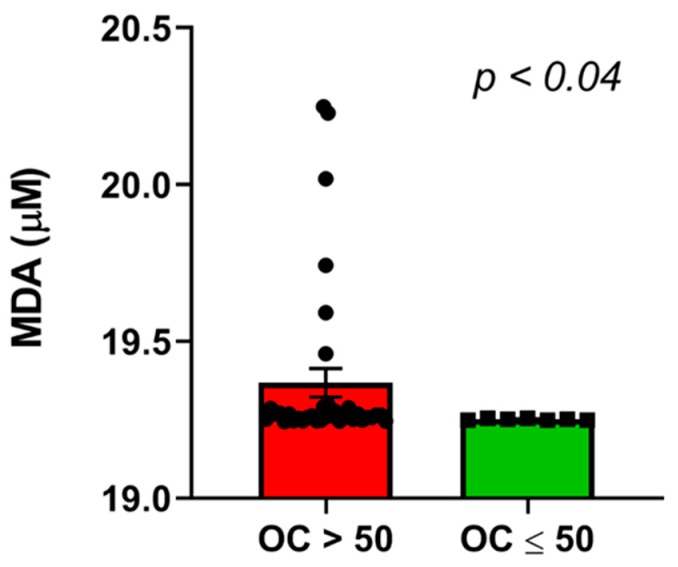
Graphical representation comparing malondialdehyde level in high and low overcommitted nurses. Malondialdehyde (MDA) was measured in the urine samples of the study participants and compared (OC > 50 vs. OC ≤ 50; *p* < 0.04) using Mann–Whitney test.

**Figure 7 ijerph-16-03243-f007:**
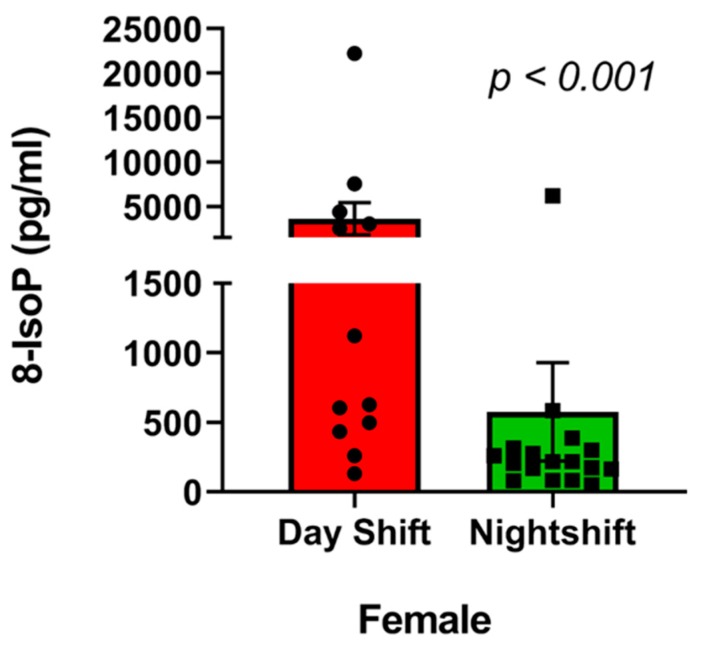
Graphical representation comparing 8-isoprostane levels in day shift and night shift female nurses. 8-isoprostane (8-IsoP) levels were measured in the urine samples and compared (female day shift nurses vs. female night shift nurses; *p* < 0.001) using Mann–Whitney non parametric test. A significant difference was observed in 8-IsoP levels between female day shift and night shift nurses (*p* < 0.001).

**Table 1 ijerph-16-03243-t001:** Descriptive statistics of the study participants.

Characteristics	Total (−*n* = 42) *−n* (%) or Mean (SD) *	Median (Range)
Age (years) *	33.04 (9.79)	30 (22–60)
Sex		
Female	29 (69)	
Male	13 (31)	
BMI *	28.2 (6.36)	27.3 (19.1–44.6)
Marital status		
Married	21 (50)	
Single	21 (50)	
Years of experience		
≤10	33 (79)	
>10	9 (21)	
Shift work		
Day shift	16 (38)	
Rotating shift	26 (62)	
Weekend work		
Yes	259 (71.2)	
No	9 (2.5)	
Work hours/week		
≤40	21 (50)	
>40	21 (50)	

* Standard deviation (SD) values for Age and Body Mass Index (BMI).

**Table 2 ijerph-16-03243-t002:** Characteristics of exposures (ERR, OC) and outcomes (oxidative biomarkers) in the study participants.

Characteristics	Total (−*n* = 42) *−n* (%) or Mean (SD) *	Median (Range)
ERR		
Low ≤ 1	3 (7)	
High > 1	39 (93)	
OC		
Low ≤ 50	7 (17)	
High > 50	35 (83)	
MDA * (µm)	19.34 (0.249)	19.35 (19–20)
8-IsoP * (pg/mL)	1452.9 (3682.7)	349.68 (51–22248)
Antioxidant * (µm)	480.2 (0.0007)	480.4 (479–481)

* Standard deviation (SD) values for Malondialdehyde (MDA), 8-Isoprostane (8-IsoP)and Antioxidant.

**Table 3 ijerph-16-03243-t003:** Normality test (Shapiro–Wilk test) showing distribution of job stress indicators and oxidative stress biomarkers.

Variables	Test Statistic	*p* Value
ERR	0.98	0.78 (Normal)
OC	0.95	0.14 (Normal)
MDA	0.45	<0.0001
8-IsoP	0.38	<0.0001
Antioxidant	0.92	<0.01

**Table 4 ijerph-16-03243-t004:** Levels of oxidative stress biomarkers in participants with high and low ERR (Wilcoxon–Mann–Whitney test).

Variables	ERR < 1 −*n* (3) Mean, Median, SD	ERR > 1 −*n* (39) Mean, Median, SD	Test Statistic	*p* Value
MDA	19.42, 19.27, 0.27	19.34, 19.25, 0.25	78.5	0.50
8-IsoP	183.71, 83.62, 176	1550.58, 352.024, 3807	32	0.11
Antioxidant	480.1, 480.49, 0.75	480.21, 480.44, 0.73	55	0.66

**Table 5 ijerph-16-03243-t005:** Levels of oxidative stress biomarkers in participants with high and low OC (Wilcoxon–Mann–Whitney test).

Variables	OC < 50		OC > 50	
	−*n* (7) Mean, Median, SD	−*n* (35) Mean, Median, SD	Test Statistic	*p* Value
MDA	19.25, 19.25, 0.002	19.36, 19.26, 0.26	89	0.03 *
8-IsoP	1113.13, 376.82, 1393	1520.91, 316.89, 3998	168	0.56
Antioxidant	480.36, 480.65, 0.82	480.17, 480.41, 0.71	177	0.38

* *p* < 0.05; significant.

**Table 6 ijerph-16-03243-t006:** Demographic and socioeconomic indicators as risk factors for high ERR.

Characteristics	OR (95% CI)	*p* Value
Age (years)		
20–30	1.00 (Reference)	
>30–40	1.084 (0.051–23.208)	0.70
>40	0.423 (0.005–33.97)	0.63
Sex		
Female	0.640 (0.051–7.95)	
Male	1.00 (Reference)	0.72
Years of experience		
≤10	1.976 (0.05–69.20)	
>10	1.00 (Reference)	0.70
Shift work		
Day shift	1.00 (Reference)	
Rotating shift	0.309 (0.031–3.09)	0.31
Weekend work		
Yes	2.913 (0.033–254.75)	
No	1.00 (Reference)	0.63
Work hours/week		
≤40	1.00 (Reference)	
>40	0.818 (113-5.922)	0.84
BMI		
Obese (≥30)	2.653 (0.206–34.15)	0.56
Overweight (25.0–29.9)	1.731 (0.208–14.39)	0.95
Normal (<25)	1.00 (Reference)	

**Table 7 ijerph-16-03243-t007:** Demographic and socioeconomic indicators as risk factors for high OC.

Characteristics	OR (95% CI)	*p* Value
Age (years)		
20–30	1.00 (Reference)	
31–40	1.903 (0.111–32.70)	0.95
>40	4.143 (0.107–160.47)	0.48
Sex		
Female	2.967 (0.397–22.18)	0.28
Male	1.00 (Reference)	
Years of experience		
<10	1.00 (Reference)	
>10	0.177 (0.009–3.57)	0.26
Shift work		
Day shift	1.00 (Reference)	
Rotating shift	2.683 (0.45–15.94)	0.28
Weekend work		
Yes	1.502 (0.020–115.13)	0.85
No	1.00 (Reference)	
Work hours/week		
≤40	1.00 (Reference)	0.34
>40	2.429 (0.37–15.56)	
BMI		
Obese (≥30)	2.006 (0.226–17.77)	0.68
Overweight (25.0–29.9)	1.684 (0.223–12.727)	0.86
Normal (<25)	1.00 (Reference)	

**Table 8 ijerph-16-03243-t008:** Correlation between ERR and oxidative stress biomarkersin the study participants (Spearman rank correlation test).

Oxidative Stress Biomarkers	vs. ERR	
	Rho (ρ)	*p* Value
MDA	0.087	0.58
8-IsoP	−0.11	0.46
Antioxidant	0.106	0.50

**Table 9 ijerph-16-03243-t009:** Correlation between OC and oxidative stress biomarkers in the study participants (Spearman rank correlation test).

Oxidative Stress Biomarkers	vs. OC	
	Rho (ρ)	*p* Value
MDA	0.13	0.40
8-IsoP	−0.19	0.20
Antioxidant	0.10	0.49

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
