# Peer review of "A Cross Sectional Study Evaluating Psychosocial Job Stress and Health Risk in Emergency Department Nurses"

_ijerph, 2019, doi:10.3390/ijerph16183243_

Round 1
Reviewer 1 Report
This is a very interesting paper on the occupational health of a healthcare professional group.
Although the results do not support the study hypothesis in terms of statistical signifcance, there is ground to believe that the riskfactors studies as well as the stress biomarkers are related to increased occupational stress.
What this paper seems to be weak on, is the conclusions and the suggestions on how to deal with the problem at the workplace setting. There are strategies and methods to manage such situations and the literature has some examples to offer. It would be a good idea to enrich the paper with some additional references.
Finally, some minor language editing is required, e.g lines 56, 71, 109, 149, 229 and 347.
Reviewer 2 Report
Thank you for allowing me to review this manuscript. This study was a cross sectional study of psychosocial stress in nurses (n=-50)
Why effort reward imbalance questionnaire. has this scale been validated in the health care/acute care setting such as the ED?
Other aspects of shift work has been associated with development of cardiac disease such as short sleep or sleep dysregulation. Did you collect any data on this?
Other covariates such as years of experience (older more “battle hardened” staff may tolerate stress better or worse) you state that majority of staff were 10 years or less (79%), do we have an idea of that as being in the ED setting or other less acute settings?
Am curious why the authors chose to focus on oxidative stress markers compared to other more common ways to operationalize cardiac risk in stress such as autonomic system dysfunction, HPA axis assessments, HR variability etc. I think it’s an innovative take on this interesting topic
